# Anatomical and Biomechanical Properties of the Junction between Stem and Aerial Roots of *Selenicereus undatus*

**DOI:** 10.3390/plants12132544

**Published:** 2023-07-04

**Authors:** Bennett Pauls, Thea Lautenschläger, Christoph Neinhuis

**Affiliations:** Institute of Botany, Department of Biology, Technische Universität Dresden, 01216 Dresden, Germany

**Keywords:** Cactaceae, *Hylocereus undatus*, anatomy, failure, X-ray tomography, epiphytes

## Abstract

Cacti have a distinct adaptation to arid conditions with a massive water storing tissue surrounding a weak central woody cylinder. However, they have not been studied as extensively as other plants have been. *Selenicereus undatus* is a hemi-epiphytic root climber that attaches itself to supporting plants or rocks with adventitious roots. The anatomy and biomechanics of the adventitious roots were studied using light microscopy, X-ray tomography and pullout and uniaxial tensile tests. The central cylinder of the roots is highly lignified and is connected to the vascular system of the shoot in a peculiar way. Xylem elements of the root turn 90 degrees towards the base of the shoot and merge laterally and below the junction with those from the shoot. Tensile and pull-out tests showed that failure occurs either at the root or junction, with the fracture surface mainly comprising the area where xylem elements from the root merge with those from the shoot. However, damage to the cortical tissue was minimal, and the measured forces showed that adventitious roots have a high safety factor. Even a complete failure of the junction after pullout does not result in severe injury to the cortex, which could lead to water loss or the entry of pathogens.

## 1. Introduction

The (hemi)epiphytic life form is generally applied to plants structurally depending on other plants as substrate. Roughly 9% of vascular plants are considered to be epiphytes although the exact use of the term is not always straightforward. There is an accepted distinction between holo-epiphytes that spend their whole life cycle on the supporting plant and hemi-epiphytes that root in the soil at least during some growth stages. Additionally, the terms “nomadic vines” and “aerial hemiparasites” have been proposed for plants such as lianas and mistletoes, respectively [1,2,3,4]. All these habits have in common that they need some kind of anchorage to their host such as winding, root climbing or, in the case of some Cactaceae, spines [5,6]. In the case of attachment via roots, distinct anatomical specializations can be found, depending on whether the roots have an anchoring, nutrition conducting or combined function. This results in differences in for example vessel formation, lignification or stele shape, as described for Araceae [7,8,9].

*Selenicereus undatus* (Haw.) D.R.Hunt (syn. *Hylocereus undatus*) is a root-climbing species amongst a small number of (hemi-) epiphytic Cactaceae [1] (Figure 1). It grows up to a height of 10 m in trees or on rocks with highly branched shoots [10,11]. The shoots are three-ribbed with a diameter of 4–7.5 cm. Flowers and branches always originate from areoles on the ribs whereas aerial roots originate from the area in between (Figure 1B). The genus originally occurs in Central and South America, but is nowadays a widely cultivated crop plant in South East Asia producing fruits known as dragon fruit, pitaya orpitahaya [12,13]. Only few biomechanical and anatomical data of stems of climbing cacti are available so far, e.g., for the recently studied *Selenicereus setaceus* [5,6]. The biomechanics and ontogeny of cacti junctions have been studied for branches in *Opuntia ficus-indica*, *Cylindropuntia bigelovii* and *H. undatus* as well as for flowers in *H. undatus* [14,15]. For the aerial roots that are produced abundantly along the shoots, only cursory anatomical studies exist [16,17]. This investigation aims to provide a more detailed understanding of the aerial root anatomy, mechanical properties and failure modes especially of the junction between root and shoot.

## 2. Results

### 2.1. Anatomy and Morphology

Adventitious roots are produced as early as during the growth of individual shoot segments and usually only on the sides facing a supporting surface or on the downward-pointing surfaces in the case of horizontally growing shoots. The roots are covered with a brownish papery bast sheath, which is connected to the shoot surface but does not extend into the cortex (Figure 2C).

The cross-section of the aerial roots (Figure 2A) shows a characteristic arrangement of tissues starting with a rather prominent lignified pith in the center followed by a star-shaped xylem, a vascular cambium, and finally the bast comprising alternating parenchymatous cells and lignified fibers. The groups of parenchymatous cells between the xylem poles most probably represent at least partly the primary phloem. The lignified tissue extends radially to form a star-shaped structure composed almost exclusively of wood fibers (Figure 2B). These bundles are separated by ray-like tissues, that remain parenchymatous, but were observed to be lignified as well in some younger roots (Figure 2D, black arrows). Sometimes, a differentiation in the xylem can be observed between an inner and outer layer, differing in the density of the tissue representing the primary and secondary xylem (Figure 2B). Vessel elements were observed in a low number in some samples (Figure 2D, red arrows), while they seemed to be lacking in others (Figure 2A,B). Longitudinal sections of a mature root showed the existence of perforated vessel elements (see Appendix A). Phloroglucinol HCl staining confirmed the occurrence of lignification throughout the root cross-section (see Appendix A). Additional cross-sections of different roots show that the general arrangement of the anatomical elements is always the same, but details can vary, for example the number of xylem bundles, xylem shape or pith shape (see Appendix A).

X-ray tomography revealed how the fibers and conducting tissue of the xylem run from the shoot’s central cylinder into the root (Figure 3). It was possible to segment the two parts of the root xylem, with the inner part highlighted in green and the outer highlighted in yellow, most probably representing the primary and secondary xylem of the central cylinder of the root. Figure 3A shows a 3D model of the xylem network in the vicinity of the junction. It has to be noted that the differentiation between inner and outer xylem bundles is not as precise inside the junction where it intersects with the shoot xylem. The xylem elements of the root all bent down sharply by about 90 degrees towards the base of the shoot, even on the adaxial side (Figure 3B). Subsequently, they fused with existing xylem elements of the shoot along a distance of about 2 cm. Fibers on the adaxial side have only minor direct connections to the xylem running towards the shoot apex. Interestingly, in the area of root attachment and fusion with the shoot, the shoot xylem forms a kind of depression, as if the root itself is formed at very early stages of shoot development, deeply anchored in the primary xylem, involving also the inner parts of the wood cylinder (Figure 3B), while the later formed secondary shoot xylem does not contribute to the linking of root and shoot (Figure 4). A comparison of X-ray and light microscopic sections can be found in the Appendix A.

### 2.2. Biomechanics

In the tensile tests performed with rather straight individual sections of the adventitious roots, failure always occurred along their free length as brittle fracture. In the pull-out tests, two distinct kinds of failure were recognized. Either the roots failed along the free root length as in the tensile tests, or the junction itself failed. In the latter case, a portion of the shoot xylem was pulled out and remained attached to the root. Figure 5A shows the pulled out root with attached parts of the shoot’s wood cylinder. Figure 5B shows the equivalent fracture surface of the shoot segment from which the root has been pulled out. The failure on the shoot‘s basal side of the root occurred further away from the junction than on the shoot’s apical side corresponding to the area in which the xylem of the root merges with that of the shoot, as described above. Despite the rather large damage to the wood cylinder, only minor damage to the parenchymatous water storing cortex and the epidermis was recognized. The almond-shaped xylem section comprises an area almost as wide and thick as that of the shoot xylem and is about two centimeters in length; however, only a hole corresponding to the root diameter is visible in the epidermis (Figure 6). Preliminary tests showed that the outer bast layer of the root is only loosely connected to the central xylem elements and therefore is only of very limited importance to junction strength. The diameter of the root does not correlate with the cross-section of a pulled out junction (Figure 7A).

The maximum recorded force as the onset of failure was used as a strength indicator. It was measured at a median of 8.42 N for root failure and 13.52 N for junction failure during pull-out tests as well as 11.85 N for failure in the tensile tests with individual root sections. Normalization to breaking stress gave respective median values of 3.45 MPa, 6.11 MPa and 10.78 MPa as well as Young’s moduli of 40.31 MPa, 63.09 MPa and 255.86 MPa. The corresponding statistical indicators are given in Table 1 and Table 2, and the corresponding plots are shown in Figure 7B,C.

## 3. Discussion

*Selenicereus undatus* relies as a hemi-epiphytic root climber on a safe attachment to the support. As a highly branched species climbing up to 10 m on trees and on rocks it can reach a considerable weight and therefore is at risk of falling down. The three possible modes of failure include the following events: (1) individual roots lose contact to their substrate, which we do not consider here; (2) root breakage occurs in between shoot surfaces and the point of attachment to the substrate; or (3) in the worst case, pull-out of the root occurs from the shoot at the junction. The latter failure mode could eventually cause massive damage to the xylem and the water-storing primary cortex, and, as a result, attack by pathogens.

Generally, obtaining entire unfragmented sections of root material is difficult. Although adhesive tape helps to stabilize the root during cutting and staining, the connection between the pith and xylem is fragile and fragments easily, as does the cambial area between the xylem and bast. Through a top view of a cross-section of a root, however, all parts were observed to be connected.

The cross-section exhibited the general arrangement of tissues characteristic of aerial roots, with a central cylinder enclosing a pith, small amounts of phloem located at the end of rays separating xylem poles and a primary cortex presumably covered by a rhizodermis in early stages of development. The latter, however, is already replaced very early on by secondary tissues, including the bast, that most likely forms as a kind of sheath acting as a protective barrier against desiccation during root growth. This would also explain why it is found only on the root part extending outside of the shoot but not within the shoot cortex. These tissues were dead in almost all samples, with collapsed parenchymatous tissue alternating with fibers, comprising a structure similar to the bark structure described for *Selenicereus setaceus* shoots [6]. A similar root structure has been reported for the below-ground roots of other Cactaceae, for example *Ferocactus acanthodes*, *Pilosocereus pachycladus* and *Opuntia basilaris*, but with an unlignified pith [18,19,20,21].

The separation of the inner and outer fiber bundles, clearly visible in the X-ray tomography sample (Figure 3B), can also be found in the anatomical sections, although is less pronounced (Figure 2B) and most probably represents the border between the primary and secondary xylem. The striking separation in the tomography picture is likely due to the preparation process, which involved maceration and subsequent drying. The inner bundles shrank together with the root pith while the outer bundles were more resistant against desiccation.

It remained unclear from the roots studied whether the lignified rays between the xylem poles in younger roots (Figure 2D, black arrows) always exist during a distinct growth stage or whether this is just a result of the development of individual roots. As far we studied, no mature root had any lignified tissues between xylem poles resulting in a higher flexibility of the root, as previously described for lianas (see below). Vessel elements, visible as wider structures (Figure 2D, red arrows), could only be observed in some of the roots. As the whole mature root is lignified, it seems to be specialized towards material strength, not water conduction, and is responsible for anchorage only. It is interesting to note that in the study on *S. setaceus*, the whole shoot was modified according to the mechanical needs from almost self-supporting basal parts to searchers looking for new support in tree crowns or hanging over rocks [5,6]. It may be possible that the differences in lignification of the tissue between the xylem poles represent similar adaptations, but so far this cannot be substantiated based on our data.

However, in *S. undatus*, the general structure of the root is highly modified such that the pith is fully lignified, and the xylem shows secondary growth, but not growth resulting in a closed ring. Instead, the secondary xylem is subdivided into individual segments, virtually consisting of fibers only, separated by parenchymatous tissue.

From a mechanical point of view, the aerial roots can be compared to a tensioned cable with the pith acting as the core and the fibers as the wire strands, although the fibers are actually not winding around the pith [22]. Far-better-studied plant structures with a similar purpose and tissue arrangement include that of lianas [3,23,24]. *Clematis maritima* and *Aristolochia macrophylla* show a similar anatomy albeit consisting of individual vascular bundles separated by wood rays [25,26].

The X-ray tomography and pulled out roots show that the root is deeply anchored in the wood of the stem throughout the entire wall thickness of the central cylinder (Figure 3A,B). A simplified representation of the fiber connection in the junction is shown in Figure 8.

This suggests that the initiation of adventitious root growth starts at a very early stage of shoot development since a connection to more central parts of the wood in the shoot in an advanced or fully developed state would not be possible. Later initiation of adventitious roots, when the shoots were fully developed, could not be observed in the studied specimen cultivated in the greenhouse. This could either have been a result of the growing conditions (the shoots did not have contact to soil or a suitable substrate for attachment) or a characteristic of the species. Considering the variable number of roots per shoot, a mechanism to start the root growth has to exist, for which factors still have to be identified. Based on the limited significance of the observations in the greenhouse, adventitious roots seem to be produced primarily on shoots that grow upwards or horizontally, while they are lacking on shoot sections that hang down. The complete lignification of the central cylinder including the pith and the way the fibers are connected to the shoot point towards a primary function of the roots as anchoring structures, although observations in the literature suggest the possibility of a nutritional function if the soil is reached [17]. The results obtained from the studies in *S. setaceus* could be interpreted in the same way [5], although the question is not specifically addressed.

The difference in breaking force for root failure in pull-out and tensile tests and junction failure can be explained from a structural point of view. In the tensile test, the force acts along the fiber orientation, i.e., the direction in which the highest tensile load can be applied. The differences in the recorded forces in the two different test modes are mostly explained by the fact, that in the tensile tests, selected short, straight segments of the root were used, while in the pull-out tests, the roots showed various amounts of curvature. These roots were straightened in the beginning, which most certainly caused pre-damage to the material, and therefore failure at lower forces.

In the pull-out test, the force was transferred through the junction to the fibers running along the axis of the central cylinder which were then stressed perpendicularly to their preferred direction. The shape of the fracture surface is due to the difference in fiber connection to the central cylinder. On the shoot’s abaxial side, the fibers are only bent around one corner. On the adaxial side, the fibers are led back in a basal direction and thus around two corners, which increases the stresses acting in the junction. In addition, these fibers have only few connections to fibers of the wood cylinder running further up the shoot. There is almost no lateral connectivity to the fibers in the shoot xylem so acting forces are only transmitted along one axis but not around its circumference as it was observed in the connection between the shoots and flowers or branches via the areoles [14]. As a result, fibers in the adaxial side of the junction break directly at their bending and connections points while those on the abaxial side are peeled off the shoot xylem together with the root and eventually break further down.

The maximum recorded forces for failure do not take into account the different individual values for the diameter or cross-sectional area of the central cylinder of the roots. However, this is the most suitable way to test the biological function of the mechanics of root attachment. The breaking stress allows a comparison of the tensile tests and those pull-out tests in which the root failed, but not of those where the junction failed. Here, the fracture surface is irregularly shaped and the failure mode different, not allowing to compare Young’s moduli or breaking stress adequately. It should be generally noted that the calculation result of the fracture surface area from pictures is an underestimation, as the fracture surface is not flat but domed, and that the root base radius is an overestimation of the radius of the mechanically relevant cross-section, as the bast adds a non-negligible thickness. Still, the breaking stress calculated with the root base diameter showed significant differences between pull-out and tensile tests. These should follow the same explanation as that for the differences between the breaking forces. The difference in Young’s modulus is primarily caused by a structural difference between the samples in the pull-out and tensile test. In the pull-out test, the forces act on the junction first and thus on fibers running perpendicular to the force direction. In the tensile test, the forces act parallelly to the fiber direction.

The force necessary for a failure, either along the root or in the junction, shows that the aerial root attachment is developed in such a way that it possesses a high safety factor, which is common in plants [27]. The average shoot of about 50 cm length weighs around 150 g, and a mature fruit weighs 350–500 g [14]. A single root would be able to hold the whole shoot and fruit under tensile load. This does not take into account that there are multiple roots per shoot that can anchor to a substrate and that the shoot itself has at least some self-supporting capabilities. It would be more likely that the shoot–substrate connection fails, which was not part of the current study.

The pull-out test indicated a lower breaking force for root failure than that for junction failure, in opposition to the tensile tests. Since root failure in both cases should follow the same mechanism, the explanation for the difference most likely is pre-damage during the straightening of curved roots. However, even if the root is pulled out of the shoot, causing considerable damage to the wood cylinder, the damage in the primary cortex of the plant including the epidermis is surprisingly limited to the diameter of the root. There seems to be no strong connection between the root material and the cortical cells of the shoot, which limits the damage to the shoot xylem, minimizing water loss from a wound or the probability of pathogens penetrating the cortical tissue.

Important aspects still need to be studied to understand the mechanisms of the aerial roots of *S. undatus*. These are the exact processes of root initiation and early development towards a non-nutritional root on a cellular level and how the roots anchor themselves to their substrate as well as the exact mechanism through which damage is avoided in the cortex and epidermis of the shoot. For a better general understanding of Cactacea anatomy, the aerial roots should also be compared to below-ground roots.

## 4. Materials and Methods

### 4.1. Plant Material

Plant material was taken between February and May 2022 from a plant cultivated at the Botanical Garden of Technische Universität Dresden, Dresden, Germany, in the desert greenhouse under seasonal conditions (Figure 1A). The plant (IPEN CU-0-DR-009517) grows freely on trellises to a height of roughly 4 m with pronounced branching. Terminal shoots of different ages with developed aerial roots were cut and processed while fresh within hours or stored in ethanol (70%). We investigated only aerial roots not in contact with the substrate since we were interested in the properties of the root itself and the junction to the shoot xylem, but not in the mode of attachment to a substrate.

### 4.2. Anatomy and Morphology

All shoot samples were measured in length and weighed before further processing. The sampled shoots weighed between 2–3 g/cm and had a length of 15–60 cm. The number of fully developed aerial roots per shoot section was highly variable; some sections had none, while others had more than 20, with a diameter of 0.7–2.7 mm measured just above the shoot surface and a length of 30–200 mm.

For the preparation of root cross-sections, shoot sections with roots were cut and embedded in polyethylenglycol 4000 (Merck KGaA, Darmstadt, Germany). The parenchymatous ribs had to be cut to fit into the embedding container. All root sections were prepared from root parts from in between 5 to 20 mm measured from the shoot surface. Sectioning was carried out on a sliding microtome (M. Schanze, Leipzig, Germany) with a slice thickness of 10–30 µm. To stabilize the fragile sections, the block surface was covered in adhesive tape (Scotch Magic Tape, 3M Deutschland GmbH, Neuss, Germany) before each cut [28,29,30]. Sections were bleached in sodium hypochlorite solution (2.8%) and subsequently stained with carmine–methyl green (Carl Roth GmbH, Karlsruhe, Germany) which stains lignified tissues blue-green and non-lignified tissues purple [31]. Lignification was also confirmed via staining with Phlorogluciol HCl. The stained sections were photographed using a light microscope (VHX-970F, Keyence AG, Osaka, Japan) with an integrated camera (VHX-970F, CMOS-image sensor, Keyence AG, Osaka, Japan).

For X-ray tomography, shoot samples were macerated in cold water for roughly one week until the central cylinder and woody roots could be isolated via careful rinsing. The central cylinder was subsequently airdried for several days, then finish-dried at 60 °C in a drying cabinet (Heraeus T12, Heraeus Instruments, Hanau, Germany) and sawed into sections of ~15 mm length with one aerial root each. Tomography measurements were carried out on ProCon CT-XPRESS (ProCon X-ray GmbH, Sarstedt, Germany) and analyzed using the software Dragonfly 3.5 (Object Research Systems Inc., Montreal, QC, Canada). Segmentation was performed manually. Additional images of macerated samples were taken on a Zeiss Supra 40 VP scanning electron microscope (Carl Zeiss Microscopy Deutschland GmbH, Oberkochen, Germany).

### 4.3. Biomechanical Testing

Tensile tests were performed on a total of 11 isolated aerial root samples and pull-out tests were performed on a total of 52 roots still connected to the shoot, using a universal testing machine and corresponding software (Zwick/Roell Allround Line Z005, testXpert II V3.5, Zwick/Roell GmbH & Co. KG, Ulm, Germany). Only fully developed roots, as determined according to the dried-out root tips, were used. For pull-out, tests the free root ends were glued onto paper strips with cyanoacrylate glue (Silisto Sekundenkleber, mfi Metall + Fastening Industrie GmbH, Twist, Germany) to prevent slipout from the clamps during testing. Subsequently, the root was lead through a 20 mm hole in a fixed aluminum plate of a 10 mm thickness to ensure that the junction was not influenced by clamping forces during testing. The sample diameter was measured using digital calipers (Precise PS 7215, Burg-Wächter, Wetter-Volmarstein, Germany) at the sample’s center or at the contact point of the root and shoot surface for tensile and pull-out testing, respectively. The free sample length was chosen to be 40 mm, the pre-load was chosen to be 0.1 N and the testing speed was chosen to be 2 mm/min. The force was applied in line with the root. The resulting elongation was recorded from the traverse movement and the resulting force was recorded with a 5 kN load cell (Zwick/Roell xforce P, Zwick/Roell GmbH &Co. KG, Ulm, Germany). The software OriginLab 2021 Pro (OriginLab Corporation, Northampton, MA, USA) was used to calculate characteristic biomechanical values and compare them statistically using a Kruskal–Wallis ANOVA and Dunn’s test at a significance level of 0.05. For the sake of comparison, the root cross-section was used to calculate all normalized values. The cross-sectional area of pulled out junctions was calculated with the software ImageJ (National Institutes of Health, Bethesda, MD, USA) from pictures taken after testing.

## Figures and Tables

**Figure 1 plants-12-02544-f001:**
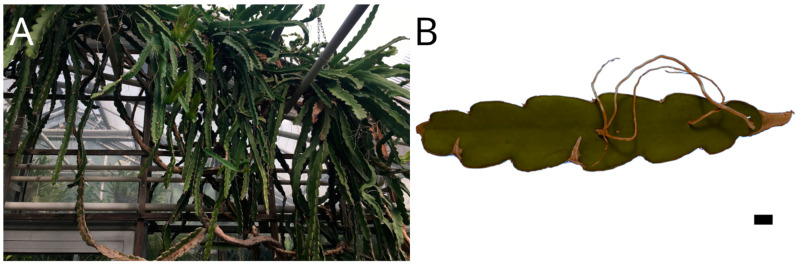
(**A**) Highly branched mature individual of *Selenicereus undatus* growing on trellises in the Botanic Garden of TU Dresden. (**B**) Shoot section of *S. undatus* with aerial roots. Scale: 10 mm.

**Figure 2 plants-12-02544-f002:**
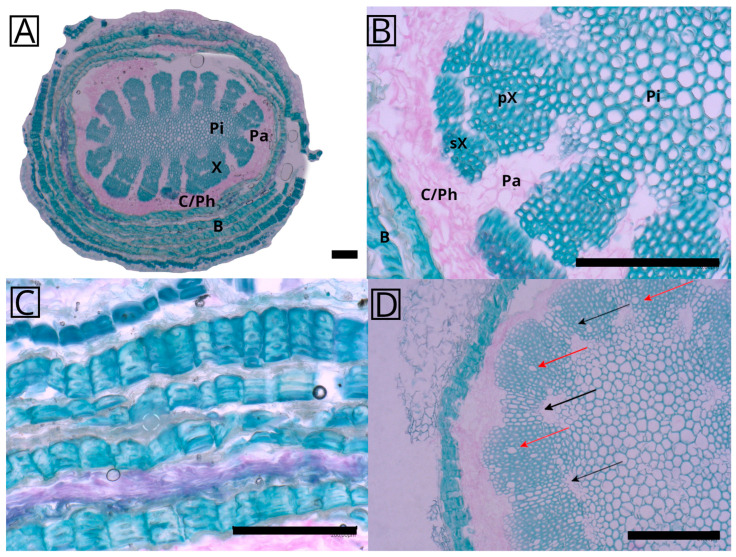
Stained cross-section of the aerial root of *Selenicereus undatus*. (**A**) Lignified tissues in turquoise, and parenchymatous non-lignified tissues in pink. The xylem poles are directly connected to the lignified pith and separated by a parenchymatous tissue. A ring of meristematic/parenchymatic tissue surrounds the xylem followed by a protective bast sheath consisting of alternating layers of collapsed parenchymatous tissue and fiber cells. (**B**) Detail of root cross-section, showing two individual xylem poles connected to the pith and separated by parenchyma. The pith is completely lignified, and the xylem poles can be differentiated into primary and secondary xylem by the respective density and the visible separating border. (**C**) Closeup of the bast sheath with alternating layers of sclerenchyma fibers and collapsed parenchymatous tissue. Pith direction to the bottom. (**D**) Cross-section of a younger root with partly lignified tissue (black arrows) connecting xylem poles and vessel elements (red arrows). The bast sheath is considerably thinner than that in the older roots consisting of only one layer of fibers. C: vascular cambium, Ph: phloem, Pa: parenchymatous tissue, X: xylem poles, pX: primary xylem, sX: secondary xylem, Pi: pith. Scale: 200 µm.

**Figure 3 plants-12-02544-f003:**
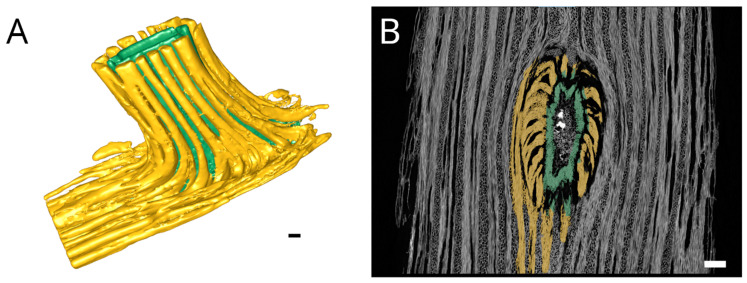
(**A**) 3D model of the fibers of the root-shoot-junction. Inner fiber bundles of the root are marked in green, outer ones are marked in yellow, and the shoot base is towards the lower left. While the inner xylem parts are connected to more central parts of the shoot xylem, the outer xylem elements are primarily connected to the peripheral parts of the shoot xylem. (**B**) Segmented tangential slice of X-ray tomography of central cylinder with aerial root. Inner parts of the root xylem are marked in green, outer ones are marked in yellow, the shoot base is towards the bottom. It becomes clearly visible that all elements of the root xylem sharply bend downwards to merge with the shoot xylem beneath the junction. Above the junction only limited connections occur. Scale: 1 mm.

**Figure 4 plants-12-02544-f004:**
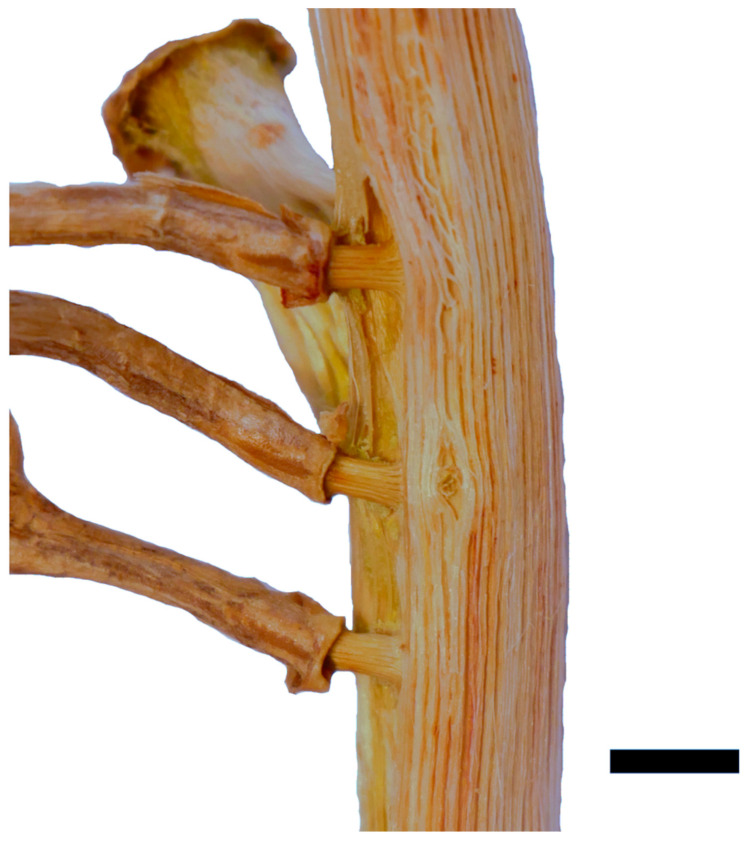
Macerated central cylinder with connected aerial roots. The cylinder is indented around the junction, and the bast sheath of the roots does not extend to the junction. Scale: 5 mm.

**Figure 5 plants-12-02544-f005:**
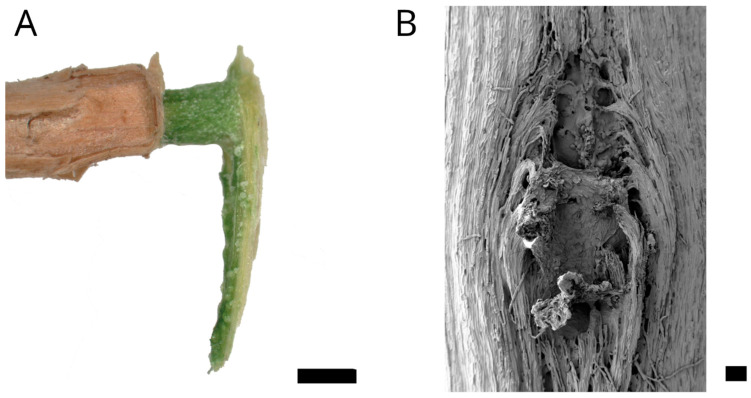
(**A**) Adventitious root after pull-out test. Clearly visible is the attached part of the shoot xylem that corresponds to the area in which the root xylem and shot xylem merge and which has been completely extracted from the shoot. Apparent is the brown bast sheath, formed just above the shoot epidermis, which is lacking inside the primary cortex. The shoot’s basal side is toward the bottom. Scale: 1 mm. (**B**) SEM image of the fracture surface in the central cylinder, macerated after the pull-out test. Failure occurred directly at the point of attachment. The fibers of the shoot xylem bend around the junction and the failure points of individual xylem parts of the root are clearly visible. Scale: 200 µm.

**Figure 6 plants-12-02544-f006:**
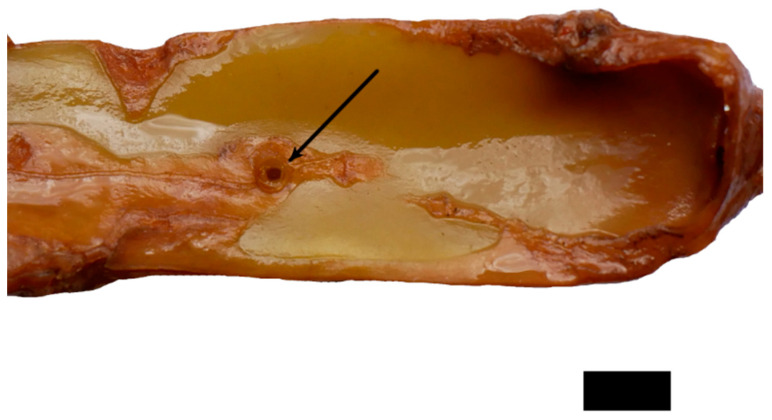
Fixed sample after pull-out test showing the hole left behind by the root (black arrow). There is no further damage visible on the shoot surface. Scale: 5 mm.

**Figure 7 plants-12-02544-f007:**
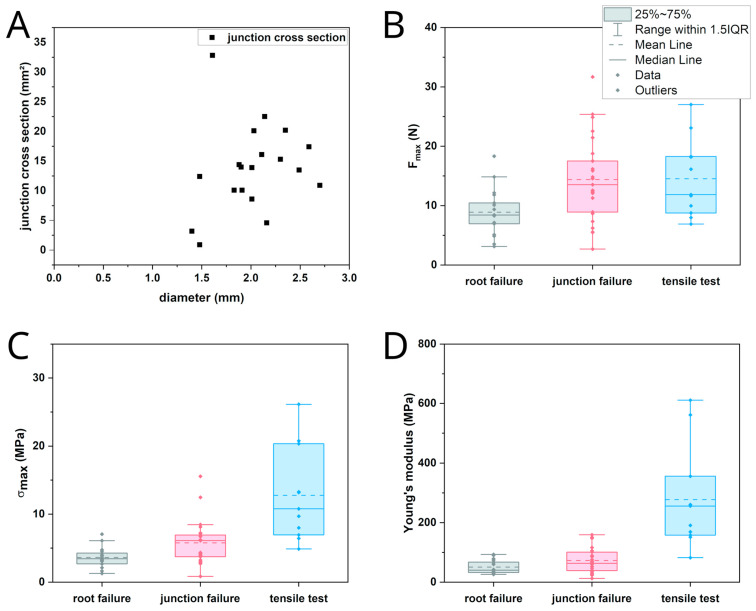
Overview of the results of biomechanical testing. (**A**) Influence of the root base diameter on the pulled out junction area. (**B**–**D**) Comparison of breaking force in N (**B**), breaking stress in MPa (**C**) and Young’s modulus in MPa (**D**) of the different failure types. The box represents data between the first and third quartile. Whiskers include data points within 1.5× of the interquartile range. The median is marked with a solid line and the mean is marked with a dashed line.

**Figure 8 plants-12-02544-f008:**
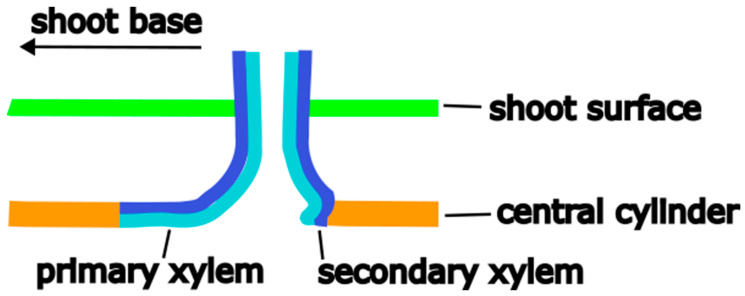
Schematic representation of the aerial root junction in a longitudinal section showing the difference in connection of the fibers on root sides.

**Table 1 plants-12-02544-t001:** Median values of biomechanical parameters. Interquartile range and sample size (n) are given in brackets; associated plots shown in Figure 7.

	Breaking Force (N)	Breaking Stress (MPa)	Young’s Modulus (MPa)
root failure	8.42 (5.11, n = 17)	3.45 (1.74, n = 17)	40.31 (38.42, n = 17)
junction failure	13.52 (9.35, n = 25)	6.11 (3.55, n = 25)	63.09 (64.39, n = 25)
tensile test	11.85 (9.96, n = 11)	10.78 (13.41, n = 11)	255.86 (196.25, n = 11)

**Table 2 plants-12-02544-t002:** *p*-Values from Dunn’s tests after Kruskal–Wallis ANOVA. Associated plots are shown in Figure 7, significance level is 0.05, and significant differences are marked with *.

	Breaking Force (N)	Breaking Stress (MPa)	Young’s Modulus (MPa)
root failure–junction failure	0.01693 *	0.06606	0.54207
root failure–tensile test	0.07513	6.55195 × 10^−6^ *	4.04988 × 10^−6^ *
junction failure–tensile test	1	0.00632 *	1.86102 × 10^−4^ *

## Data Availability

Data are contained within the article or Appendix A.

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
