# Peer review of "Anatomical and Biomechanical Properties of the Junction between Stem and Aerial Roots of Selenicereus undatus"

_plants, 2023, doi:10.3390/plants12132544_

Round 1
Reviewer 1 Report (Previous Reviewer 2)
The authors have fully addressed my comments.
Author Response
Thank you for taking the time to review our paper.
Reviewer 2 Report (New Reviewer)
Interesting research on the relationship between plant structure and function, which makes further investigations possible and necessary for a complete understanding
Results:
Lines 80-82: „These bundles are separated by ray like tissues, that remains parenchymatous, but were observed to be lignified as well in younger roots.” - In my opinion these anatomical differences can be worthy of further investigations that focus on the possible different forces affecting the roots. Or do these roots have the same starting point and direction of growth?
The other possible explanation for the different xylem structure is that since the distance from the root tip of the sections is different (on the basis of lines 319-320), the degree of differentiation - due to secondary thickening - is also different.
Figs. 3.: I really miss a stem cross-section at the root exit. Without it, the stem-root trachea connections are not clear to me.
Legend of Figs.3.: Please add that: “Inner fiber bundles of root are marked in green, outer in yellow, shoot base is towards the lower left. While the inner xylem parts are connected to more central parts of the shoot xylem, the outer xylem elements are primarily connected to the peripheral parts of the shoot xylem. (B) Segmented tangential slice of X-ray tomography of central cylinder with aerial root. Inner parts of the root xylem marked in green, outer in yellow, shoot base towards the bottom.”
Please, sign in Fig.2.B and D the „outer-”and „inner fibres” of Figs. 3.!
Figure 4. is beautiful!
Line 122: „failure occurred always along their free length as brittle fracture.” Please explain me what the “free length” means in this sentence?
Lines 132-136: These sentences suggest that when the root breaks out, the damage remains below the parenchyma and epidermis layers, because these tissues did not break through. Is it true?
Line 155: „The maximum recorded force (please, show here the data) as the onset of failure was used as a strength indicator.”
Figs. 7B,C,D: The stars indicating significances in figures 7.B,C,D are not consistent with the significances in the table 2.
Line 182: at the end
The discussion is a well-written section that partly answers some of my questions and comments that I asked earlier. (For example lines 200-212). However, I am missing a drawing model that would summarize the forces that act on the mechanical tissues / fibers of such a root while performing its function. According to the results of biomechanical tests do the two types of fibers and the lignified pith help the root function?
Material and methods:
Line 330: “shoot samples were macerated in water” – It was hot water or cold water? Sample was boiled or no?
Round 2
Reviewer 2 Report (New Reviewer)
The Supplementary Figs. 4C,D are very informative and nice. It would be worth showing them in the article as parts of Figs. 3, 5, or 8.
This manuscript is a resubmission of an earlier submission. The following is a list of the peer review reports and author responses from that submission.
Round 1
Reviewer 1 Report
The article deals with anatomy and biomechanics of aerial roots of pitaya. The subject is potentially interesting. However, the article suffers from several major flaws. First of all, if I understand correctly, only one plant was tested. This is utterly unacceptable if it’s so. Second, the anatomy is insufficiently described and thus the conclusions might be entirely wrong. In addition, tomography results are probably an artefact. Specific comments are below.
Title- I propose to add “pitaya” to the title- “Anatomical and Biomechanical Properties of the Junction between Stem and Aerial Roots of Selenicereus undatus (pitaya)”
I do not fully understand the distinction in your text between climbers and epiphytes, as per your definitions they overlap. Please elaborate more.
Please write more about aerial roots and their anatomy in the Intro- especially about their ability to root when reaching ground.
L28- the terms nomadic vines and aerial hemiparasites have been proposed- please change to
the terms “nomadic vines” and “aerial hemiparasites” have been proposed
M&M- Please transfer this part after the introduction.
How may biological samples (from different plants) did you test?
Fig. 2 – The anatomy of Cactaceae roots is unusual, such large pith is usually found in stems. It’s an interesting feature. Could you please write a bit on that in the text (with relation to the relevant literature) and possibly also provide a longitudinal section?
Are all the roots shown in the stage of secondary growth? From the pith to the bast, everything is a transporting tissue. There’s no cortex or endodermis, for instance. Are there younger stages of development you can show?
The tissue shown as cambium looks like cambium and phloem to me. Please indicate phloem clearly.
Also, in the discussion you write (L172) “The cross section exhibited the general arrangement of tissues characteristic for aerial roots, with a central cylinder enclosing a pith, small amounts of phloem in between xylem poles and a primary cortex presumably covered by a rhizodermis in early stages of development. The latter, however, is already replaced very early by secondary tissues, the bast, that most likely forms as a kind of sheath acting as a protective barrier against desiccation during root growth”- so which one is phloem? In the figure you marked the tissue between the poles as parenchyma, not as phloem, and the bast is supposedly the phloem. It’s very confusingly written and presented.
L63: “c: cambium, Pa: parenchymatous tissue, x: xylem poles, Ph: pith.”- please change everything either to capital letters or small letters, be consistent. Transfer this part to the end of the legend. Add letters to panel A as well. Also, in this figure the abbreviations are not customary, and must be changed. “Ph” is usually “phloem”, not pith- and it’s quite confusing. Please change to “P” or “Pi”.
The claim the mature root does not have conducting elements is very strange. First of all, without longitudinal sections you cannot really claim that- very probably there are tracheids or very narrow vessels, not everything there is fibers. A longitudinal section must be provided.
Lignification, figure 2D- I’m not sure that what we are seeing is indeed lignified tissue. TBO is not a very reliable stain in terms of cell wall component colour. It’s true that this hue is often indicative of lignification, but it can indicate other components as well. I would advice to do a more precise staining. For instance, Phloroglucinol HCl is a very good stain for lignin. It’s a chemical reaction producing pink colour, and thus would not stain any other components.
Fig 7- the axis titles should be formatted to bold and to larger font sizes. Panel A does not have a border. The panels are too far from each other. Please make this figure more presentable.
Tomography- the results presented are likely a consequence of drying artifact. The tissue was dried at 60 degrees, which is quite aggressive and would cause shrinkage and damage to structure. The experiment should be repeated with CPD dried samples.
Reviewer 2 Report
Review of MSC titled "Anatomical and biomechanical properties of the junction between stem and aerial roots of Selenicereus undatus " by Bennett Pauls , Thea Lautenschläger , Christoph Neinhuis et al., submitted to Plants (plants-2303547).
The authors present an analysis of the junction between stem and adventitious roots (aerial type), including a description of the root’s and junction's anatomical features and biomechanical properties. The main conclusion was that investigating anatomical features and mechanical measurements confirmed better adaptation of roots to anchorage than water uptake.
In my opinion, the authors should have added two pieces of information to clarify the studying object.
1. The authors should have provided information on what distance from the root tip the cross-sections were made (near the junction or middle part of the root).
2. I wonder if roots at a comparable stage of development were used in the mechanical tests (e.g. with a well-developed bast sheath fig 2 A, C or immature stage presented in Fig. 2D, and how the differences (if any) could imply the mechanical test results. There is some information dealing with the length and diameter of roots only.
Minor points
Line 72 --clarify cork cambium or vascular cambium
Line 79 – how do you determine the degree of lignification
Fig 4 and Fig 5, some description makes the images more clearly ( bast sheath, root xylem, shoot epidermis)
Fig 7 mean value in the plot is not marked with a square as written in the caption
Lines 177-178 “This would also explain why it is found only outside the shoot but not within the shoot cortex”. Do you mean shoot or root in this sentence?
Round 2
Reviewer 1 Report
The study is based on one single plant specimen, thus unfortunately entirely unacceptable for publication. It’s not a rare plant, and the authors could have obtained more specimens if they wanted….
Author Response
The concerns regarding the number of plants used should have been sufficiently adressed in our previous review response.